# OpenReview forum: "Uncovering Competing Poisoning Attacks in Retrieval-Augmented Generation"
_ICLR.cc/2026/Conference — Submitted to ICLR 2026_

### Official Review · Reviewer_WSXS · 2025-10-24

**Soundness:** 4
**Presentation:** 4
**Contribution:** 2
**Rating:** 4
**Confidence:** 4

**Summary:**

The authors identify a novel and yet interesting research topic, which defines the competition among attackers of the same RAG system. A Bradley-Terry model is considered to formulate the success of attackers. Thorough experiments have been conducted, including single-attacker settings, competition settings, as well as their counterparts under the presence of recent defense methods. The results show interesting and unconventional results, where single attackers may not work well at the presence of other attackers. Furthermore, a new benchmark is proposed for future evaluations.

**Strengths:**

A novel research question is identified. The experimental results are interesting. Thorough numerical studies are conducted, including those under recent defenses.

**Weaknesses:**

However,  I identify several weakness:

(1) Although the problem of having multiple attackers is truly realistic, it is hard to formulate the problem into a research setting. In other words, while I appreciate the authors' attempt and great effort, several practical problems may still exist (and can hardly be formulated or assumed away). For example, it is totally understandable that simulations have to require all attackers have the same level of capability. In practice, however, some attackers may have higher level of access to the corpus, or higher rates of injecting new information into the database, or the capability to access the retriever and hence inject more valuable and likely retrieved documents, or the algorithm to identify high-value queries dynamically and more accurately.

(2) The novelty of the proposed method is rather weak. The novelty mainly lies in the novel research topic and the construction of the new benchmark. The execution of the idea, on the other hand, mainly relies on existing methods, including existing RAG attack algorithms and the Bradley-Terry method for attacker ability evaluation. If this is presented as a benchmark paper, it should be fine, though.

**Questions:**

What does it mean high-value queries? Practically how do we identify them?

It might be helpful to introduce briefly what the two datasets are, and why and how they can be used to evaluate the proposed attacker competition problem.

What's the baseline RAG system on which the attacks are compared? What's the retriever?

---

> ### Author Response · Authors · 2025-11-13
> **Response to Reviewer WSXS's concerns [Part 1]**
>
> Thank you for the valuable time and effort you have dedicated to reviewing our paper. We would first like to express our sincere appreciation for your recognition of our work, including your acknowledgement that the research problem is novel, the experiments are sufficiently thorough, and the empirical findings are insightful. Regarding the concerns and limitations you raised, we hope the following responses will address them.
>
> ## **1 Response to W1&Q1**
> >W1: (1) Although the problem of having multiple attackers is truly realistic, it is hard to formulate the problem into a research setting. In other words, while I appreciate the authors' attempt and great effort, several practical problems may still exist (and can hardly be formulated or assumed away). For example, it is totally understandable that simulations have to require all attackers have the same level of capability. In practice, however, some attackers may have higher level of access to the corpus, or higher rates of injecting new information into the database, or the capability to access the retriever and hence inject more valuable and likely retrieved documents, or the algorithm to identify high-value queries dynamically and more accurately. \
> >Q1: What does it mean high-value queries? Practically how do we identify them?
>
> ***Response***: **[For Weakness1]** We fully agree with your perspective. This problem is indeed challenging to model without introducing a gap from real-world scenarios. We also acknowledge that some assumptions may have been simplified or omitted during our modeling process, which we discuss in Appendix E (Limitation) and Appendix F (Future Work). In reality, the environment reflects a complex game-theoretic setting rather than a simple competitive one. Nevertheless, we believe this remains a highly valuable research problem. Competitive attacks are both plausible and inevitable, and our intention in this work is to make an initial, incremental contribution that provides useful insights for future studies. We see this as a step toward a more complete understanding and resolution of the problem, and we will continue to advance this line of research in subsequent work.
>
> **[For Weakness1]** Secondly, our considerations align with yours: we also aim to analyze an attacker from multiple dimensions rather than solely from attack outcomes. Our analysis is presented in Appendix D (Attack Taxonomy), where we systematically evaluate factors such as the attacker’s access to the corpus or retriever, computational overhead, and other relevant dimensions. In addition, to provide further insights for future research, we also conducted a preliminary game-theoretic analysis examining the systemic advantages introduced by different attack orders. The corresponding experiments and analysis are presented in Appendix C.5 (Attack Order). Furthermore, we analyze what an attack method should prioritize when being designed to achieve a stronger competitive advantage. This systematic examination is provided in Appendix C.6 (Trigger or Content?).
>
> **[For Question1]** High-value queries refer to queries of substantial importance—those for which attackers have strong incentives to manipulate the system. The term does not have a strict or universally accepted definition. Examples include political persuasion during a presidential election, competitive dynamics between Coca-Cola and Pepsi, or competition among different retailers selling the same product during Black Friday. Broadly speaking, these are queries with high stakes—economic value, political influence, public-opinion impact—and involve multiple parties with conflicting interests. In practice, we can often infer whether a query is likely to be targeted by multiple attackers by analyzing system-level traffic patterns, such as identifying what topics the public is currently focused on or what events are trending. These signals help determine which queries are more likely to attract coordinated or competing malicious interference.

---

> > ### Author Response · Authors · 2025-11-13
> > **Response to Reviewer WSXS's concerns [Part 2]**
> >
> > ## **2 Response to W2**
> >
> > >w2: (2) The novelty of the proposed method is rather weak. The novelty mainly lies in the novel research topic and the construction of the new benchmark. The execution of the idea, on the other hand, mainly relies on existing methods, including existing RAG attack algorithms and the Bradley-Terry method for attacker ability evaluation. If this is presented as a benchmark paper, it should be fine, though.
> >
> > ***Response***: Yes. The primary contribution of our work is to formally articulate a problem that has been largely overlooked in the literature yet occurs frequently in real-world settings: competitive attacks. Our focus is on modeling this problem, analyzing the limitations of existing approaches, and proposing a completely new evaluation framework. In this sense, our work introduces both a new research problem and a new benchmark for studying it.
> >
> > ## **3 Response to Q2**
> >
> > >Q2: It might be helpful to introduce briefly what the two datasets are, and why and how they can be used to evaluate the proposed attacker competition problem.
> >
> > ***Response***: Thank you very much for raising this question. \
> > **[Which Dataset?]** Our experiments are primarily conducted on the NQ[1] and MS[2] datasets, and we additionally evaluated the mMARCO[3] dataset to ensure that our conclusions hold across different languages. A detailed description of all datasets and preprocessing procedures can be found in Appendix B.1 (Data Preparation), where we provide a comprehensive discussion.
> >
> > **[Why NQ, MS and mMARCO?]** The reason we selected these two datasets is that, in existing academic research—not only in studies on Poisoned RAG, but also in work on QA tasks, RAG tasks, and retrieval tasks—they are among the most widely adopted benchmarks. These datasets are broadly recognized by the community as high-quality and are collected from real-world data sources, making them reliable and representative for evaluating both attacks and defenses.
> >
> > ## **4 Response to Q3**
> >
> > >What's the baseline RAG system on which the attacks are compared? What's the retriever?
> >
> > ***Response***: Thank you very much for raising this question. The full RAG setup used in our experiments is described and discussed in detail in Appendix B.5 (Implementation Details), including choices of RAG parameters, model configurations, and other implementation considerations. The retriever we adopt is Contriever[4], which is widely used in many RAG pipelines. The rationale behind this choice, along with further explanations, is provided in Appendix B.2 (LLMs and Retriever).
> >
> > ----
> > **Reference**
> >
> > [1] Natural Questions: A Benchmark for Question Answering Research. TACL 2019 \
> > [2] MS MARCO: A Human Generated MAchine Reading COmprehension Dataset. NIPS 2016 \
> > [3] mMARCO: A Multilingual Version of the MS MARCO \
> > [4] Unsupervised Dense Information Retrieval with Contrastive Learning
> >
> > ----
> >
> > Finally, we would like to once again express our sincere appreciation for your valuable suggestions and for the time and effort you devoted to evaluating our work. We hope that our responses have addressed your concerns. If you have any further questions, please feel free to let us know—we will do our utmost to resolve them. If there are no additional concerns, we kindly and sincerely ask you to consider raising your evaluation of our work, as it is truly important to us. Thank you very much.

---

> ### Author Response · Authors · 2025-11-27
> **Looking Forward to Your Response**
>
> **Dear Reviewer WSXS**: \
> Thank you very much for the valuable suggestions you provided for our work. As the discussion period is approaching its end, we apologize for commenting again to ask whether our responses have fully addressed your concerns. We provide a brief summary of our responses below to help clarify how we addressed your concerns.
> 1. **W1:** We acknowledge the gap between our simplified setting and real-world game-theoretic dynamics, which we discuss in Appendices E–F. **Still, competitive attacks are realistic and important, and our work offers an initial step toward understanding them.** We also evaluate attackers beyond outcome metrics: Appendix D provides a multi-dimensional taxonomy, while Appendices C.5–C.6 analyze attack order and the design factors—such as trigger vs. content—that shape competitive advantage.
> 2. **[Q1: What does it mean high-value queries? Practically how do we identify them?]**: High-value queries are those with strong incentives for manipulation—high stakes in politics, economics, or public opinion—such as elections, brand competition, or major sales events. The term has no strict definition, but such queries typically attract multiple adversarial interests. In practice, system-level signals like trending topics or traffic spikes help identify which queries are most likely to face coordinated or competing attacks.
> 3. **[W2: Novelty Concern]**: Our work introduces the problem of competitive attacks, identifies limitations in existing evaluations, establishes a more principled assessment framework, and analyzes the essential strengths and weaknesses of different attack designs to provide insights for future research.
> >If this is presented as a benchmark paper, it should be fine, though.
>
>     In this sense, our work introduces both a new research problem and a new benchmark for studying it.
> 4. **[Q2: Dataset Details]**: We conduct experiments mainly on NQ and MS, with additional evaluation on mMARCO to test cross-lingual robustness. Details on datasets and preprocessing are provided in Appendix B.1. These benchmarks are widely used across QA, RAG, and retrieval research, offering high-quality, real-world data and broad community acceptance, making them well suited for evaluating attacks and defenses.
> 5. **[Q3: What's the baseline RAG system on which the attacks are compared? What's the retriever?]**: The full RAG setup used in our experiments is described and discussed in detail in Appendix B.5 (Implementation Details), including choices of RAG parameters, model configurations, and other implementation considerations. The retriever we adopt is Contriever[4], which is widely used in many RAG pipelines. The rationale behind this choice, along with further explanations, is provided in Appendix B.2 (LLMs and Retriever).
>
> Finally, we would like to once again express our sincere appreciation for your valuable suggestions and for the time and effort you devoted to evaluating our work. **We greatly appreciate your recognition, and we likewise believe this is an important yet underexplored research problem.** We hope that our responses have addressed your concerns. If you have any further questions, please feel free to let us know—we will do our utmost to resolve them. If there are no additional concerns, we kindly and sincerely ask you to consider raising your evaluation of our work, as it is truly important to us. Thank you very much.

---

### Official Review · Reviewer_cCQE · 2025-10-31

**Soundness:** 1
**Presentation:** 2
**Contribution:** 2
**Rating:** 2
**Confidence:** 5

**Summary:**

This paper presents the concept of competing attacks, a multi-adversary threat model for retrieval-augmented generation systems where multiple attackers attempt to influence the same query at the same time.

**Strengths:**

A novel competing attack is proposed to target the RAG system.

**Weaknesses:**

1. The threat model assumed in the paper is unrealistic.

2. The runtime of the proposed attack and all baseline methods is not evaluated.

3. Allowing five poisoned documents to be injected into the system per query is impractical.

4. The paper overlooks several recent attack and defense approaches.

**Questions:**

1. The paper assumes that the attacker either has access to a proxy retriever or can observe the retriever’s output, which is unrealistic. In real-world scenarios, it is nearly impossible for an attacker to obtain such information. Moreover, the authors’ statement that “the strength of the attack assumption is not the central focus of our study” is inappropriate. For an attack paper, the proposed method must demonstrate effectiveness under realistic threat models, as practicality and realistic assumptions are essential to evaluating attack feasibility.

2. Some attack methods, such as GASLITE, benefit from more advanced optimization mechanisms, while simpler baselines are under-optimized. This imbalance compromises the fairness of the comparisons.

3. Simulating pairwise competitions among all attackers across datasets is computationally expensive, yet the paper does not provide any runtime or efficiency analysis of the proposed framework.

4. The experimental setup allows five poisoned documents per query to be injected into the RAG system. This configuration is unrealistic because, as shown in [a][b], the number of truly relevant texts among the top-5 retrieved documents per query is typically fewer than five (e.g., in the NQ dataset). As a result, the number of poisoned documents exceeds the number of relevant ones, which artificially inflates the attack success rate. A more practical setting would restrict the attacker to injecting only one poisoned document per query.

5. Several recent and more advanced poisoning attacks on RAG systems, such as [a][c], are not included in the comparison. The authors should evaluate their method against these stronger and up-to-date baselines to demonstrate competitiveness.

6. The range of defenses examined in the paper is narrow. Additional robust defenses, such as [d][e], should be incorporated.


[a] Practical Poisoning Attacks against Retrieval-Augmented Generation.

[b] Benchmarking Poisoning Attacks against Retrieval-Augmented Generation.

[c] FlippedRAG Black-Box Opinion Manipulation Adversarial Attacks to Retrieval-Augmented Generation Models.

[d] SafeRAG Benchmarking Security in Retrieval-Augmented Generation of Large Language Model.

[e] Traceback of Poisoning Attacks to Retrieval-Augmented Generation.

---

> ### Author Response · Authors · 2025-11-13
> **Response to Reviewer cCQE's concerns [Part 1]**
>
> Thank you for the valuable time and effort you have dedicated to reviewing our paper. We sincerely appreciate your suggestions; ***however, it appears that there is a misunderstanding about the nature of our work. The key misunderstanding is that we do not propose a new attack method. Instead, our contribution is to introduce a new attack problem.*** In response to the concerns you raised, we hope to clarify them through the following additional explanations.
>
> ## **1 Response to W1&Q1**
> >W1: The threat model assumed in the paper is unrealistic. \
> >Q1: The paper assumes that the attacker either has access to a proxy retriever or can observe the retriever’s output, which is unrealistic. In real-world scenarios, it is nearly impossible for an attacker to obtain such information. Moreover, the authors’ statement that “the strength of the attack assumption is not the central focus of our study” is inappropriate. For an attack paper, the proposed method must demonstrate effectiveness under realistic threat models, as practicality and realistic assumptions are essential to evaluating attack feasibility.
>
> ***Response***: Regarding the concern about the strength of our threat model, we provide detailed clarification in Section 2 THREAT MODEL and Appendix A.3 THREAT MODEL OF COMPETING POISONING ATTACK.
> Importantly, the core contribution of our paper is not to propose a new poisoning attack, but to introduce a new attack problem.
>
> As for why we adopt the assumption that “the attacker either has access to a proxy retriever or can observe the retriever’s output, which is unrealistic,” our rationale is the following:
> the currently evaluable poisoning methods in the literature all rely on this assumption. We agree that such access does not reflect practical deployment settings; however, as you noted in Questions 5 and 6, a proper evaluation should include more attack methods and more defense methods. To support this broader evaluation, our threat model must encompass the shared assumptions of existing attacks—otherwise, only a very small subset of methods would be eligible for evaluation.
>
> Finally, we emphasize that the relative strength or weakness of the assumption does not affect the existence of competing poisoning attacks as an objective phenomenon. Our purpose is singular:
> to evaluate as many attack methods as possible so that the resulting conclusions are robust and generalizable.
> If permitted, we could restrict the evaluation to threat models that match real-world constraints more closely, but currently only a few attack methods operate under such realistic assumptions, which would prevent us from obtaining solid and comprehensive conclusions.
>
> ## **2 Response to Q2**
> >Some attack methods, such as GASLITE, benefit from more advanced optimization mechanisms, while simpler baselines are under-optimized. This imbalance compromises the fairness of the comparisons.
>
> ***Response***: Regarding this point, we are not entirely sure about your intended meaning. The contribution of our work is precisely to evaluate the strengths and weaknesses of different methods under competitive poisoning. If an attack method adopts a better optimization strategy, it should indeed achieve better results.
> This is analogous to comparing the translation capabilities of ChatGPT and BERT: we cannot claim that such a comparison is “unfair” simply because BERT has a simpler architecture or is easier to train. The comparison focuses on which model performs better on the translation task, and architectural complexity is irrelevant to the validity of the comparison.
>
> ## **3 Response to W2&Q3**
> >W2:The runtime of the proposed attack and all baseline methods is not evaluated.\
> >Q3: Simulating pairwise competitions among all attackers across datasets is computationally expensive, yet the paper does not provide any runtime or efficiency analysis of the proposed framework.
>
> ***Response***: Regarding this concern, running 1k evaluation rounds in PoisonArena is essentially equivalent to performing 1k RAG queries, without introducing any additional computational or storage overhead. All computations can be executed on a single RTX 3090 GPU (larger models such as Phi-4 may require a higher-end GPU).
> The computational cost depends solely on the attack method itself and is independent of our evaluation protocol. The attack is executed first, and only then is the evaluation performed.
> A detailed analysis of the computational overhead for each attack method is provided in Appendix D: ATTACK TAX. ***In summary, the cost of running one evaluation round is essentially identical to that of performing a single RAG query, with no additional computational overhead introduced.***

---

> > ### Author Response · Authors · 2025-11-13
> > **Response to Reviewer cCQE's concerns [Part 2]**
> >
> > ## **4 Response to W3&Q4**
> > >W3: Allowing five poisoned documents to be injected into the system per query is impractical. \
> > >Q4: The experimental setup allows five poisoned documents per query to be injected into the RAG system. This configuration is unrealistic because, as shown in [a][b], the number of truly relevant texts among the top-5 retrieved documents per query is typically fewer than five (e.g., in the NQ dataset). As a result, the number of poisoned documents exceeds the number of relevant ones, which artificially inflates the attack success rate. A more practical setting would restrict the attacker to injecting only one poisoned document per query.
> >
> > ***Response***: In our main experiments, we follow the standard practice of injecting five poisoned documents[1][2][3][4][5], as adopted by the majority of recent work. As you mentioned, paper [c] also injects five, and **even ten**, poisoned documents. It is important to clarify that “injecting five documents” means adding five poisoned entries into a database containing tens of thousands of documents (e.g., in our NQ and MS datasets, the corpus size exceeds 50k). In this setting, the retriever must select at most five poisoned documents out of ~50k candidates, which is a reasonable and realistic difficulty level.
> >
> > Regarding your suggestion of testing scenarios where only one or two poisoned documents are injected, we share the same motivation. Accordingly, we performed a systematic exploration in Appendix C.7 HYPERPARAMETER INFLUENCE. As shown in Figure 14, we restricted each attack method to inject only 1, 2, 3, 4, or 5 poisoned documents. The results are fully consistent with our main findings, indicating that our conclusions are not sensitive to the number of injected poisoned documents.
> >
> > ## **5 Response to W4&Q5&Q6**
> > >W4: The paper overlooks several recent attack and defense approaches. \
> > >Q5: Several recent and more advanced poisoning attacks on RAG systems, such as [a][c], are not included in the comparison. The authors should evaluate their method against these stronger and up-to-date baselines to demonstrate competitiveness.\
> > >Q6: The range of defenses examined in the paper is narrow. Additional robust defenses, such as [d][e], should be incorporated.
> >
> > ***Response***: We fully agree with your point that obtaining more robust conclusions requires evaluating more methods. However, the attack and defense methods included in our paper already represent the most advanced and representative techniques we were able to identify at the time of completing this work (covering black-box, white-box, prompt-injection, and adversarial-optimization–based approaches).
> >
> > The papers you referenced as [a] and [c] are indeed works we closely followed. However, they did not release their code, making reproduction and direct comparison infeasible. We even contacted the authors (including those of paper [2]) as early as April this year, and they replied that the work was still incomplete and therefore the codebase could not be shared.
> >
> > In addition, the paper you referred to as [e] is an attack attribution method rather than a defense method; it only identifies the source or cause of an attack, but does not provide an actual defense mechanism.
> >
> > ----
> > **Reference**
> >
> > [1] PoisonedRAG: Knowledge Corruption Attacks to Retrieval-Augmented Generation of Large Language Models. USENIX Security Symposium 2025 \
> > [2] “Glue pizza and eat rocks” - Exploiting Vulnerabilities in Retrieval-Augmented Generative Models. EMNLP 2024 \
> > [3] Prompt Perturbation in Retrieval-Augmented Generation based Large Language Models. KDD 2024 \
> > [4] Human-Imperceptible Retrieval Poisoning Attacks in LLM-Powered Applications. ASE 2024 \
> > [5] AGENTPOISON: Red-teaming LLM Agents via Poisoning Memory or Knowledge Bases. NeurIPS 2024 \
> > [6] InstructRAG: Instructing Retrieval-Augmented Generation via Self-Synthesized Rationales. ICLR 2025
> >
> > ----
> >
> > Finally, we would once again like to thank you for the valuable time and effort you have devoted to reviewing our work. We hope that our responses have addressed your concerns. If you have any further questions, please feel free to let us know—we will do our best to clarify them. Given that most of the issues you raised stemmed from certain misunderstandings (which we have made every effort to clarify), we hope we can now reach a consensus. If there are no additional concerns, we kindly and sincerely ask you to consider raising your evaluation of our work, as it is truly important to us. Thank you very much.

---

> > > ### Comment · Reviewer_cCQE · 2025-11-27
> > >
> > > All my concerns have been addressed, so I have decided to increase my score.

---

> > > > ### Author Response · Authors · 2025-11-27
> > > >
> > > > Thank you very much for your response. We are delighted that our replies have addressed all of your concerns, and we truly appreciate that your evaluation of our work has improved. We are sincerely grateful for the valuable suggestions you offered and for the time and effort you devoted to our submission. Thank you for your recognition of our work.

---

### Official Review · Reviewer_kQAJ · 2025-11-01

**Soundness:** 2
**Presentation:** 2
**Contribution:** 2
**Rating:** 4
**Confidence:** 3

**Summary:**

This paper studies competing poisoning attacks on RAG systems, cases where multiple adversaries try to push conflicting misinformation into the same query. The authors show that methods strong in single-attacker settings often fail or flip rankings when attackers compete. They introduce PoisonArena, a benchmark with new metrics (m-ASR, m-F1, and a competitive coefficient) to measure attack strength under competition. Results show that real-world RAG security needs multi-attacker evaluation, not just single-attacker tests.

**Strengths:**

1. This paper is the first to study competing poisoning attacks in RAG systems, moving beyond the oversimplified single-attacker assumption.
2. It introduces PoisonArena, a well-structured evaluation framework for multi-attacker experiments.
3. This paper provides code, simulation details, and convergence criteria for reliable replication.

**Weaknesses:**

1.Although the paper presents the idea of competing attackers as a realistic scenario, likw political misinformation, product competition, the experiments remain entirely synthetic. All poisoned documents and queries are generated using LLMs rather than derived from real user-generated or adversarial data. This disconnect weakens the paper’s central claim that competing attacks mirror real-world threat. there is no evidence that such multi-party interference actually emerges in open systems.
2. The proposed simulation framework requires repeated pairwise competitions among multiple attackers across large datasets. This design is computationally heavy, yet the authors provide no discussion or measurements of runtime, memory cost, or scaling behavior. Without efficiency evaluation, it is unclear whether PoisonArena can scale beyond small experimental settings or be adopted for large-scale security testing.
3. One of the paper’s most interesting findings that "weaker single-attacker methods outperform stronger ones under competition "is treated descriptively, not analytically. The authors offer no clear theoretical reasoning or model of interaction that explains why this inversion occurs. Without such grounding, the finding risks being dataset or setup-specific rather than a general phenomenon.
4. The setting allows up to five poisoned documents per query gives each attacker excessive influence. In most RAG pipelines, the retriever only surfaces one or two truly relevant passages among the top-k results. Granting five injected documents per query likely inflates attack success rates and makes competition dynamics less representative of real conditions. A more realistic setting should restrict attackers to one or at most two poisoned documents per query.

**Questions:**

N/A

---

> ### Author Response · Authors · 2025-11-13
> **Response to Reviewer kQAJ's concerns [Part 1]**
>
> Thank you for the valuable time and effort you have dedicated to reviewing our paper. First, we would like to express our sincere gratitude for your recognition of our work, including your acknowledgment that our contribution is the first to introduce competitive attacks setting in RAG, that our evaluation framework *PoisonArena* is well-structured, and that our implementation details are clearly presented to facilitate reproducibility. In response to the concerns you raised, we hope to clarify them through the following additional explanations.
>
> ## **1 Response to W1**
> >1.Although the paper presents the idea of competing attackers as a realistic scenario, likw political misinformation, product competition, the experiments remain entirely synthetic. All poisoned documents and queries are generated using LLMs rather than derived from real user-generated or adversarial data. This disconnect weakens the paper’s central claim that competing attacks mirror real-world threat. there is no evidence that such multi-party interference actually emerges in open systems.
>
> ***Response***: Thank you very much for raising this question. We understand that your considerations mainly involve two points:
>
> 1. Since the experimental data are synthetic, is there a gap between our setting and real-world scenarios?
> 2. Is there evidence that competitive attacks would actually occur in real-world applications?
>
> We address these two considerations in order below.
>
> ### **1.1 Is there a gap between our setting and real-world scenarios?**
>
> ***Response***: Our research can only approach this issue from an academic standpoint, aiming to highlight a potential threat that may arise in real-world RAG systems. Direct access to sensitive real-world data—such as malicious competitive behaviors in e-commerce (specific attack queries or poisoned documents) or even more sensitive political conflicts—is unattainable because these datasets are strictly restricted. The synthetic nature of our data does not undermine the objective existence of the problem itself. What we need to examine is whether current attack methods can transfer or generalize to this competitive setting. In fact, existing RAG-poisoning attacks also rely on constructed datasets rather than naturally occurring real-world corpora[1][2][3][4].
>
> For this reason, we believe that although our constructed data may not perfectly replicate every detail of real-world environments, there is no substantial distributional or conceptual mismatch that would invalidate our formulation of the problem or the conclusions drawn from our experiments. The experimental design is faithful to the core dynamics of the threat model, and the insights we obtain remain reliable and informative for understanding potential risks in deployed RAG systems.
>
> ### **1.2 Evidence that competitive attacks would actually occur in real-world applications**
>
> ***Response***: Your concern essentially asks whether such attacks truly exist in real-world environments. Our conclusion is: **if current RAG poisoning attacks are realistic and meaningful, then competitive attacks must also exist; and if existing RAG attacks warrant study, then competitive attacks in RAG constitute an urgently needed research direction.** The queries that attackers seek to manipulate—such as political election topics or product ratings—almost always involve multi-party interests. If one party can launch an attack, then inevitably multiple parties can do so as well.
>
> We provide a detailed discussion of this argument in Section 5.1, *“Why Competing Attacks Matter: Do They Reflect Real-World Scenarios?”* If any doubts remain, we can illustrate this (Also in Appendix G CASE STUDIES IN REAL-WORLD SCENARIOS) with real-world examples that already exhibit the same competitive dynamics. We list them below (in accordance with the rebuttal guidelines, we have removed the direct news URLs, although all sources can be readily found via public search):

---

> > ### Author Response · Authors · 2025-11-13
> > **Response to Reviewer kQAJ's concerns [Part 2]**
> >
> > We list them below (in accordance with the rebuttal guidelines, we have removed the direct news URLs, although all sources can be readily found via public search):
> >
> > | Sector/Domain            | Multi-attacker motive & goal                                                                    | High-value queries/entry points                                      | Illustrative real-world example (publicly reported)                                                   |
> > |--------------------------|--------------------------------------------------------------------------------------------------|------------------------------------------------------------------------|--------------------------------------------------------------------------------------------------------|
> > | Politics & elections     | Rival partisan groups shape voter perception and push conflicting claims                         | "Candidate X scandal", "live poll results"                            | Competing PACs edited/seeded fact-checking pages in the 2024 U.S. election cycle                      |
> > | E-commerce/SEO           | Sellers boost their own products and bury competitors to win rankings                            | "Best SPF 50 sunscreen", "XYZ phone review"                           | Amazon and other marketplaces battling large fake-review rings                                        |
> > | Financial markets        | Short-and-distort vs. pump-and-dump teams influence share prices                                 | "Company X earnings analysis", "XYZ short thesis"                     | Coordinated "short-and-distort" campaigns against mid-cap stocks                                      |
> > | Meme-stock communities   | Retail factions hype or trash the same ticker to steer sentiment                                 | "$KRISPY price target"                                                | Opposing Reddit groups drove conflicting narratives on meme-stock tickers                             |
> > | Malware/phishing         | Crime rings hijack trending keywords to distribute malicious downloads                           | "Free VPN download", "Microsoft 0-day patch"                          | Multiple phishing kits cloning popular login pages (e.g., 0ktapus)                                    |
> > | Package registries       | Actors race to squat typo- or look-alike package names (supply-chain)                           | `requests` typos, internal lib names                                  | Dependency-confusion attacks uploading malicious NPM/PyPI packages                                    |
> > | Local services/Maps      | Fake businesses capture emergency-service leads in high-margin niches                           | "City locksmith", "24h plumber"                                       | Networks of bogus Google Maps locksmith listings competing for calls                                  |
> > | Music streaming          | Labels, promoters and bots inflate streams and suppress rivals                                   | "Chill playlist", "New Music Friday"                                  | Bot farms and AI-generated tracks jostling for top playlist slots                                     |
> > | Academic metrics         | Citation mills and "citation rings" trade references to boost metrics                            | "Google Scholar h-index"                                              | Bulk-purchased citations affecting thousands of scholar profiles                                       |
> > | Consumer reviews         | Businesses swap 5★ for themselves and 1★ for rivals to sway ratings                              | "Best pizza in [object Object]"                                       | National regulators uncovering widespread fake reviews on local platforms                             |
> > | Social-media click farms | Creators/agencies buy views, likes, comments to out-rank competitors                             | "Funny short video"                                                   | TikTok & YouTube view-bot services openly marketed to influencers                                     |
> > | Public-health info       | Pro- and anti-vaccine groups push conflicting medical claims                                      | "Vaccine side-effect truth"                                           | COVID-19 misinformation surge with multiple factions promoting opposing narratives

---

> > > ### Author Response · Authors · 2025-11-13
> > > **Response to Reviewer kQAJ's concerns [Part 3]**
> > >
> > > ## **2 Response to W2**
> > > > 2. The proposed simulation framework requires repeated pairwise competitions among multiple attackers across large datasets. This design is computationally heavy, yet the authors provide no discussion or measurements of runtime, memory cost, or scaling behavior. Without efficiency evaluation, it is unclear whether PoisonArena can scale beyond small experimental settings or be adopted for large-scale security testing.
> > >
> > > ***Response***: Regarding this concern, running 1k evaluation rounds in PoisonArena is essentially equivalent to performing 1k RAG queries, without introducing any additional computational or storage overhead. All computations can be executed on a single RTX 3090 GPU (larger models such as Phi-4 may require a higher-end GPU).
> > > The computational cost depends solely on the attack method itself and is independent of our evaluation protocol. The attack is executed first, and only then is the evaluation performed.
> > > A detailed analysis of the computational overhead for each attack method is provided in Appendix D: ATTACK TAX. ***In summary, the cost of running one evaluation round is essentially identical to that of performing a single RAG query, with no additional computational overhead introduced.***
> > >
> > > ## **3 Response to W3**
> > > >3. One of the paper’s most interesting findings that "weaker single-attacker methods outperform stronger ones under competition "is treated descriptively, not analytically. The authors offer no clear theoretical reasoning or model of interaction that explains why this inversion occurs. Without such grounding, the finding risks being dataset or setup-specific rather than a general phenomenon.
> > >
> > > ***Response***: Thank you for raising this point. Due to space limitations in the main paper, we only provided a brief explanation of this phenomenon. However, we offer a thorough and rigorous discussion in the appendix.
> > >
> > > ### **3.1 Our findings are treated analytically and general phenomenon**
> > >
> > > First, our findings are supported by extensive experimental evidence. As shown in Figures 2–4 and Tables 1–6, we varied the model, the dataset, the language, and multiple RAG parameters, and consistently observed the same conclusion across all settings. This demonstrates that our results are robust rather than incidental.
> > >
> > > ### **3.2 "The authors offer no clear theoretical reasoning or model of interaction that explains why this inversion occurs."**
> > >
> > > To further investigate this finding, we conducted an extensive analysis in the appendix, where we also provide an explanation of the underlying cause (see Appendix C.6 TRIGGER OR CONTENT?). Through a large number of controlled comparisons, our analysis shows that different attack methods implicitly prioritize different optimization objectives. For a poisoned attack to succeed, two conditions must be met simultaneously: (1) optimizing the retrieval trigger, and (2) optimizing the misleading content.
> > > Because different methods focus on these two aspects to varying degrees, they naturally exhibit different retrieval behaviors, as illustrated in Figure 2 and related results.
> > >
> > > Our conclusion is that, provided the misleading content is sufficiently strong, improving the quality of the retrieval trigger substantially increases an attack method’s competitiveness. This explains the observed reversal effect: certain methods over-emphasize misleading content (e.g., PoisonedRAG-black) but, in competitive settings, may fail to retrieve the poisoned document at all due to insufficient trigger quality.

---

> > > > ### Author Response · Authors · 2025-11-13
> > > > **Response to Reviewer kQAJ's concerns [Part 4]**
> > > >
> > > > ## **4 Response to W4**
> > > > >4. The setting allows up to five poisoned documents per query gives each attacker excessive influence. In most RAG pipelines, the retriever only surfaces one or two truly relevant passages among the top-k results. Granting five injected documents per query likely inflates attack success rates and makes competition dynamics less representative of real conditions. A more realistic setting should restrict attackers to one or at most two poisoned documents per query.
> > > >
> > > > ***Response***: In our main experiments, we follow the standard practice of injecting five poisoned documents[1][2][4][5][6], as adopted by the majority of recent work. It is important to clarify that “injecting five documents” means adding five poisoned entries into a database containing tens of thousands of documents (e.g., in our NQ and MS datasets, the corpus size exceeds 50k). In this setting, the retriever must select at most five poisoned documents out of ~50k candidates, which is a reasonable and realistic difficulty level.
> > > >
> > > > Regarding your suggestion of testing scenarios where only one or two poisoned documents are injected, we share the same motivation. Accordingly, we performed a systematic exploration in Appendix C.7 HYPERPARAMETER INFLUENCE. As shown in Figure 14, we restricted each attack method to inject only 1, 2, 3, 4, or 5 poisoned documents. The results are fully consistent with our main findings, indicating that our conclusions are not sensitive to the number of injected poisoned documents.
> > > >
> > > > ----
> > > > **Reference**
> > > >
> > > > [1] PoisonedRAG: Knowledge Corruption Attacks to Retrieval-Augmented Generation of Large Language Models. USENIX Security Symposium 2025 \
> > > > [2] “Glue pizza and eat rocks” - Exploiting Vulnerabilities in Retrieval-Augmented Generative Models. EMNLP 2024 \
> > > > [3] Prompt Perturbation in Retrieval-Augmented Generation based Large Language Models. KDD 2024 \
> > > > [4] Human-Imperceptible Retrieval Poisoning Attacks in LLM-Powered Applications. ASE 2024 \
> > > > [5] AGENTPOISON: Red-teaming LLM Agents via Poisoning Memory or Knowledge Bases. NeurIPS 2024 \
> > > > [6] InstructRAG: Instructing Retrieval-Augmented Generation via Self-Synthesized Rationales. ICLR 2025
> > > >
> > > > ----
> > > >
> > > > Finally, we would like to once again express our sincere appreciation for your valuable suggestions and for the time and effort you devoted to evaluating our work. We hope that our responses have addressed your concerns. If you have any further questions, please feel free to let us know—we will do our utmost to resolve them. If there are no additional concerns, we kindly and sincerely ask you to consider raising your evaluation of our work, as it is truly important to us. Thank you very much.

---

> ### Author Response · Authors · 2025-11-27
> **Looking Forward to Your Response**
>
> **Dear Reviewer kQAJ**: \
> Thank you very much for the valuable suggestions you provided for our work. As the discussion period is approaching its end, we apologize for commenting again to ask whether our responses have fully addressed your concerns. We provide a brief summary of our responses below to help clarify how we addressed your concerns.
> 1. **[W1: Gap between the data distribution and the real-world data distribution]**: The datasets we use are derived from real-world sources such as NQ and MS MARCO. However, for ethical considerations in academic research, the *harmful data* in our experiments is generated by large language models rather than taken directly from real user content. This design choice does not change the fundamental setting of the problem. We have listed real-world cases in Response Part 2, and we discuss this issue in detail in Section 5.1, *“Why Competing Attacks Matter: Do They Reflect Real-World Scenarios?”* as well as in Appendix G, *Case Studies in Real-World Scenarios*.
> 2. **[W2: Cost analysis]**:  The cost of running one evaluation round is essentially identical to that of performing a single RAG query, with no additional computational overhead introduced. We also provide a detailed discussion in Appendix D: *Attack Tax*. In addition, this concern was likewise raised by **Reviewer cCQE**, and our response to that point has been acknowledged and accepted by cCQE.
> 3. **[W3: Risk of Findings]**: We conducted extensive experimental analyses in the paper, varying dataset sizes, dataset languages, model families, and experimental hyperparameters. Across all these settings, the conclusions remained consistent. In addition, to further examine the robustness of these conclusions, we carried out additional analyses, which are detailed in Appendix C.6 *“Trigger or Content?”*.
> 4. **[W4: Setting Concern]**: We conducted extensive experiments and analyses in Appendix C.7 *“Hyperparameter Influence”* specifically under the setting you were concerned about, demonstrating that these factors do not affect our conclusions or findings. This concern was also raised by **Reviewer cCQE**, and our response to that point was acknowledged and accepted by cCQE.
>
> Finally, we would like to once again express our sincere appreciation for your valuable suggestions and for the time and effort you devoted to evaluating our work. We hope that our responses have addressed your concerns. If you have any further questions, please feel free to let us know—we will do our utmost to resolve them. If there are no additional concerns, we kindly and sincerely ask you to consider raising your evaluation of our work, as it is truly important to us. Thank you very much.

---

### Official Review · Reviewer_B74c · 2025-11-10

**Soundness:** 3
**Presentation:** 3
**Contribution:** 2
**Rating:** 6
**Confidence:** 2

**Summary:**

This paper introduces the concept of "competing poisoning attacks" in RAG systems, where multiple adversaries simultaneously attempt to mnipulate the same query toward different, mutually exclusive targets. Additionally, they propose PoisonArena, a benchmark framework that evaluates poisoning methods under both single- and multi-attacker settings.

**Strengths:**

S1. There is a critical gap in existing RAG security research and this paper identifies it and fulfill this gap. The competing attacks is realisti and relevant forhigh-stakes queries in politics and public health.

S2. The evaluation is comprehensive covering seven attack methods, six LLMs and multiple datasets including (NQ, MS MARCO, mMARCO).

S3. The paper is well written and easy to follow.

**Weaknesses:**

W1. The choice of 8 incorrect answers per query seems arbitrary and lacks justification

W2. The convergence criterion based on ranking stability (Equation 7) could be sensitive to the choice of r (consecutive rounds)

W3. The specific prompt templates for generating adversarial content may introduce biases

**Questions:**

Please refer to Weakness part

---

> ### Author Response · Authors · 2025-11-13
> **Response to Reivewer B74c's concerns**
>
> Thank you for the valuable time and effort you have dedicated to reviewing our paper. First, we would like to express our sincere gratitude for your recognition of our work, including your acknowledgment that the new attack setting we proposed is relevant to real-world scenarios, the evaluation method is novel, the experiments are thorough, and the writing process is well-structured. In response to the concerns you raised, we hope to clarify them through the following additional explanations.
>
> ## **1 Response to W1**
> >W1: The choice of 8 incorrect answers per query seems arbitary and lacks justification.
>
> ***Response***: In our experiments, we considered up to seven of the most advanced attack methods. As an extreme example, a highly valuable query may be attacked by many attackers using different attack methods. In our setting, this corresponds to seven attackers simultaneously targeting the same query. To implement this setting, we had to prepare sufficient incorrect answers to ensure that each attacker was randomly assigned a different attack target. Without this, there could be a situation where at least two attackers share the same attack goal, which would lead to unfair attack results. **Therefore, under the competitive setting, evaluating *n*  attack methods requires preparing at least *n* distinct incorrect answers.**
>
>
> As for why we prepared 8 incorrect answers, this was due to our concern during data preparation that the difficulty levels of the incorrect answers might vary. Therefore, we considered preparing n+1 incorrect answers, from which one would be discarded after manual verification (if all were reasonable, none would be discarded). This is why, as mentioned in Appendix B.1 (Data Preparation), we prepared a maximum of 8 answers. Of course, this approach can be extended to any number of incorrect answers—if we need to evaluate ten attack methods simultaneously, we would prepare eleven incorrect answers.
>
> ## **2 Response to W2**
> >W2: The convergence criterion based on ranking stability (Equation 7) could be sensitive to the choice of r (consecutive rounds)
>
> ***Response***: Thank you very much for raising this valuable question. In our experiments, we recommend setting ( r = 50 ), which is entirely based on empirical observations. From our experiments (as shown in Figures 4, 6, 7, and 13), we found that once the ranking stabilizes, it no longer changes. To ensure the accuracy of our experiments, we simulated a large number of rounds (for example, convergence was reached at round 1000, but we still simulated an additional 3000 rounds, as shown in Figure 13, etc., to ensure the correctness of the results). Therefore, as long as  *r* is not chosen to be too small (based on our empirical findings, it should not be smaller than 50), the results will remain stable. In our experiments, we typically chose 100 rounds and executed additional simulation rounds to ensure the accuracy of the results.
>
> ## **3 Response to W3**
> >W3: The specific prompt templates for generating adversarial content
>
> ***Response***: To address this issue, we can ensure that the approach we provide minimizes bias from two perspectives:
>
> 1. **Simulation of competition:** When simulating competition, we consider all possible permutations of incorrect answers (as discussed in Section 3.3) to eliminate unfairness in the attack results that might arise from the construction of poisoned documents or the selection of incorrect answers.
>
> 2. **Uniformity of misleading information in attack methods:** For all attack methods that require misleading information, we use the same model and the same prompt construction. Even if this does not yield the optimal results, it ensures fairness across all attack methods. This is specifically detailed in Appendix B3 (Details and Alignment of Attack Methods).
>
> ----
>
> Finally, we would like to once again express our sincere appreciation for your valuable suggestions and for the time and effort you devoted to evaluating our work. We hope that our responses have addressed your concerns. If you have any further questions, please feel free to let us know—we will do our utmost to resolve them. If there are no additional concerns, we kindly and sincerely ask you to consider raising your evaluation of our work, as it is truly important to us. Thank you very much.

---

> ### Author Response · Authors · 2025-11-27
> **Looking Forward to Your Response**
>
> **Dear Reviewer B74c**: \
> Thank you very much for the valuable suggestions you provided for our work. As the discussion period is approaching its end, we apologize for commenting again to ask whether our responses have fully addressed your concerns. We provide a brief summary of our responses below to help clarify how we addressed your concerns.
> 1. **[W1: Choice of 8 incorrect answers]**: We explained this in detail in our previous response: ***to ensure a fair evaluation of all seven attack methods we consider***, the dataset must provide enough incorrect answers for meaningful comparison. Specifically, if we aim to evaluate *k* attack methods, we believe the dataset should include at least *k + 1* incorrect candidate answers.
> 2. **[W2: Convergence criterion]**: As long as *r* is not chosen to be extremely small, the results remain unaffected. Our experiments show that setting *r* greater than 50 is sufficient. This observation is supported by the patterns presented in Figures 4, 6, 7, and 13.
> 3. **[W3: Biases introduced by prompts]**:  We ensure fairness by (i) enumerating all permutations of incorrect answers during competition to remove positional or construction-induced bias, and (ii) standardizing misleading-content generation across all attack methods using the same model and prompt setup, as detailed in Section 3.3 and Appendix B.3.
>
> Finally, we would like to once again express our sincere appreciation for your valuable suggestions and for the time and effort you devoted to evaluating our work. We hope that our responses have addressed your concerns. If you have any further questions, please feel free to let us know—we will do our utmost to resolve them. If there are no additional concerns, we kindly and sincerely ask you to consider raising your evaluation of our work, as it is truly important to us. Thank you very much.

---

### Meta-Review · Area_Chair_KFxy · 2026-01-10

**Summary:**

The paper received an overall borderline rejection assessment. Reviews were mixed, with scores ranging from 2 (reject) to 6 (marginal accept). The paper introduces poisoning attacks in RAG systems and presents a useful benchmark, PoisonArena, supported by extensive empirical evaluations across multiple datasets, models, and defense settings. Noted strengths include identifying a meaningful gap beyond the standard single-attacker assumption, conducting comprehensive experiments, providing reproducible methodology, and presenting the work clearly. Primary concerns center on the realism of the threat model, the synthetic nature of the experimental data, computational cost and scalability, the lack of deeper theoretical analysis explaining the observed competitive effects, and potentially unrealistic experimental assumptions, such as the number of poisoned documents per query and uniform attacker capabilities. The paper could be strengthened by incorporating key clarifications from the rebuttal into the main text, and by proposing a more practical threat model or identifying additional realistic and actionable application scenarios.

**Reviewer Concerns:**

Concerns about the arbitrary choice of incorrect answers, convergence stability, prompt-induced bias, runtime and computational overhead, and the number of injected poisoned documents were handled with empirical justification, additional experiments, and detailed appendices. The authors also clarified misunderstandings about the paper’s goal (problem formulation and benchmarking rather than proposing a new attack), justified the use of synthetic data on ethical and practical grounds, and provided real-world analogues demonstrating that competitive attack dynamics plausibly exist.

Some concerns remain partially unresolved, primarily around external validity and realism. While the authors argue persuasively that competitive attacks are inevitable if single-attacker poisoning is realistic, direct evidence from deployed systems remains absent. Additionally, although runtime cost is argued to be comparable to standard RAG queries, the scalability of PoisonArena for very large models or continuous evaluation pipelines is not empirically demonstrated. Finally, despite improved analytical discussion, the theoretical understanding of why weaker single-attacker methods outperform stronger ones under competition is still largely empirical and may benefit from deeper formal modeling in future work.

**Reviewer Scores:**

Reviewer B74c: Likely unchanged (6 ). Concerns were addressed, but confidence was low.

Reviewer kQAJ: Likely unchanged ( 4 ). Most technical and realism concerns were answered, however some skepticism about real-world grounding may persist.

Reviewer cCQE: Explicitly improved (from 2 → higher, after Openreview Security Incident ), stating that all concerns were addressed after rebuttal.

Reviewer WSXS: Likely unchanged (4). The reviewer acknowledged the value of the research question and benchmark, with remaining concerns framed as inherent limitations.

---

### Decision · Program_Chairs · 2026-01-26

Reject